# Simulating In Vitro the Bone Healing Potential of a Degradable and Tailored Multifunctional Mg-Based Alloy Platform

**DOI:** 10.3390/bioengineering9060255

**Published:** 2022-06-15

**Authors:** Victor Martin, Mónica Garcia, Maria de Fátima Montemor, João Carlos Salvador Fernandes, Pedro Sousa Gomes, Maria Helena Fernandes

**Affiliations:** 1Laboratory for Bone Metabolism and Regeneration, Faculty of Dental Medicine, University of Porto, 4200-393 Porto, Portugal; vmartin@fmd.up.pt (V.M.); monicapgarcia@gmail.com (M.G.); pgomes@fmd.up.pt (P.S.G.); 2LAQV/REQUIMTE, University of Porto, 4100-007 Porto, Portugal; 3Research Institute for Medicines (iMed.ULisboa), Faculty of Pharmacy, University of Lisbon, 1649-003 Lisboa, Portugal; 4CQE, IMS, Departamento de Engenharia Química, Instituto Superior Técnico, University of Lisbon, 1049-001 Lisboa, Portugal; mfmontemor@tecnico.ulisboa.pt (M.d.F.M.); joao.salvador@tecnico.ulisboa.pt (J.C.S.F.); 5EST Setúbal, CDP2T, Instituto Politécnico de Setúbal, 2910-761 Setúbal, Portugal

**Keywords:** bone fixation, bio-resorbable materials, magnesium alloy, coating, functionalization, cellular response

## Abstract

This work intended to elucidate, in an in vitro approach, the cellular and molecular mechanisms occurring during the bone healing process, upon implantation of a tailored degradable multifunctional Mg-based alloy. This was prepared by a conjoining anodization of the bare alloy (AZ31) followed by the deposition of a polymeric coating functionalized with hydroxyapatite. Human endothelial cells and osteoblastic and osteoclastic differentiating cells were exposed to the extracts from the multifunctional platform (having a low degradation rate), as well as the underlying anodized and original AZ31 alloy (with higher degradation rates). Extracts from the multifunctional coated alloy did not affect cellular behavior, although a small inductive effect was observed in the proliferation and gene expression of endothelial and osteoblastic cells. Extracts from the higher degradable anodized and original alloys induced the expression of some endothelial genes and, also, ALP and TRAP activities, further increasing the expression of some early differentiation osteoblastic and osteoclastic genes. The integration of these results in a translational approach suggests that, following the implantation of a tailored degradable Mg-based material, the absence of initial deleterious effects would favor the early stages of bone repair and, subsequently, the on-going degradation of the coating and the subjacent alloy would increase bone metabolism dynamics favoring a faster bone formation and remodeling process and enhancing bone healing.

## 1. Introduction

Therapeutic approaches for bone fracture fixation and stabilization commonly rely on the use of non-degradable biomaterials with adequate mechanical properties and biocompatibility—namely metallic implants based on stainless steel, titanium, and cobalt-chromium alloys—that present a high corrosion resistance and are able to maintain long-term structural stability of the tissues [1,2]. Despite the overall adequacy, some biological limitations have been reported with long-term use, including physical irritation, chronic immuno-inflammatory activation, and the release of cytotoxic elements that may hinder biological responses and entail a second surgical procedure for implant removal, increasing morbidity, hospitalization time, and consequently healthcare costs [3,4,5]. Further, elastic modulus differences between non-resorbable metallic implants and bone tissue may induce stress shielding that can interfere with bone turnover exceling bone loss, ultimately leading to fixation deterioration that culminates with the need of a revision surgery [6]. Conventional metal alloys also impair X-ray-based imagiological diagnostic techniques, given the beam hardening and incorporated imaging artefacts [7], and report a limited use on pediatric patients, given the possible intraosseous implant migration and the potential restriction of bone growth [8]. To overcome these limitations, the use of biodegradable metal-based implants has been investigated [9]. 

In this context, magnesium-based (Mg-based) metal alloys have taken the frontline of a new generation of biodegradable metal-based materials with promising bone-related properties. For instance, these materials present similar values of density, tensile strength, and compressive yield strength relative to cortical bone [10]. Despite other properties as yield stress and elastic modulus are yet distinct, Mg-based materials are the best contender to replace the non-resorbable solutions for bone fixation, due to a proper cytocompatibility and tissue response [11], and the ability to positively stimulate angiogenesis and osteogenesis [12]. Furthermore, their relevance as temporary implant applications elapses from their natural ability to biodegrade through corrosion, in aqueous solutions, through an electrochemical phenomenon [13]. This allows a transitory effectiveness for load bearing applications, granting structural mechanical stability through the bone healing period [14].

One major concern about Mg-based materials is the control of the corrosion rate, as a fast degradation—and the associated release of hydrogen gas that can accumulate in gas pockets—can result in a premature loss of mechanical integrity and interfere with the tissue healing [15]. To overcome this problem, several techniques have been used to modulate corrosion resistance and consequently the mechanical properties of Mg-based materials, such as modifications through alloying, microstructure or grain size variations [16]; and surface treatment (e.g., anodization) or surface coating [17]. 

Previous research from our group has been focused on the development and optimization of a multifunctional coating architecture [18,19,20,21,22,23], conjoining the anodization of the Mg alloy surface with a composite coating consisting of polyether imide (PEI), diethylenetriamine (DETA), and hydroxyapatite nanoparticles (HA). The thin-film coating was found to improve the Mg alloy resistance to corrosion for more than 12 weeks, further enhancing the adhesion and proliferation of an osteoblastic cell line [18]. In vivo studies further revealed that the multifunctional coating allowed a proper control of the corrosion of the Mg alloy substrate upon implantation within a rabbit model, inducing a favorable biological response in terms of vascular ingrowth and bone osseointegration [22]. Despite the evidenced outcomes, the detailed mechanisms by which the multifunctional coating may modulate the biological response within the bone microenvironment—from a molecular and cellular perspective—are still broadly undisclosed. 

Accordingly, in the present study, a systematic and thorough assessment of the in vitro biological response to the developed materials’ extracts—considering the base Mg alloy (AZ31), the anodized surface (anodized AZ31), and the multifunctional coated surface (coated AZ31)—was conducted. Characterization was established in the most relevant cellular populations of the bone tissue microenvironment—i.e., endothelial cells and differentiating osteoblastic and osteoclastic precursor cells—that were evaluated for viability, proliferation, and functional activity parameters.

## 2. Materials and Methods

Extracts from AZ31, anodized AZ31, and anodized and coated AZ31 substrates were prepared in culture media and tested in human umbilical vein endothelial cells (HUVECs), osteoblastic differentiating human mesenchymal stem cells (hMSC) and osteoclastic differentiating human peripheral blood mononuclear cells (hPBMC). Cells were cultured in appropriate media conditions and characterized for DNA content and specific phenotype markers. The experimental protocol (Figure 1) is detailed as follows.

### 2.1. Materials’ Preparation

AZ31 samples. Coupons of commercial AZ31 Mg alloy, with 3.8 cm^2^, were mechanically polished with 2100 grit SiC paper, degreased in alcohol, and pre-treated by immersion in a 12% HF solution for 1 h.

Anodized AZ31 samples. PEO Anodization of AZ31 alloy was performed in a two-stage process: KOH 100 g/L + Na_3_PO_4_ 100 g/L (DC current 5.2 A/dm^2^, voltage up to 30 V, 10 min, 20–25 °C); KOH 6 g/L + NaF 13 g/L (DC current 3.7–1.5 A/dm^2^, voltage up to 180 V, 10 min, 20–25 °C), according to the previously reported data [24].

Coated anodized AZ31 samples. The polymeric coating was synthesized by mixing PEI in N,N-dimethylacetamide (DMAc), as a solvent, in concentrations of 15 wt.%. The mixture was stirred for 24 h at 50 °C in order to obtain a stable and yellowish but transparent solution. Diethylenetriamine (DETA) (3 wt. %) was added to this formulation. To improve the biocompatibility, nano-hydroxyapatite particles (HA) with 15–20 nm of particle size were added to the obtained polymer (2 wt. %). The composite polymeric coating (PEI+DETA+2%HA) was applied on the anodized coupons using a dip coater, following curing in an oven at 150 °C for 2 h under atmospheric conditions, obtaining a coating thickness of ~5 µm. A detailed description of the methodology was previously reported [18].

### 2.2. Surface Characterization of Material Samples

Material samples were sputter-coated with a gold palladium thin film and observed in a SEM (FEI Quanta 400FEG scanning electron microscope). 

### 2.3. Extract Preparation

Samples of AZ31, anodized AZ31, and coated anodized AZ31, with 2 cm^2^, were incubated in 10 mL of culture medium (α-MEM or M199, depending of the cell culture tested) for 24 h at 37 °C. This time point was chosen due to be the peak of the pro-inflammatory response after the occurrence of a fracture or any bone-related surgery, as well as the period of the appearance of type H vessels, which is critical for the modulation of bone formation and repair [25,26]. 

After the incubation period, the medium was collected and a series of dilutions, between 50% and 2%, was used in the cell culture experiments. The pure extracts were analyzed for the concentration of Mg and Ca ions by atomic absorption spectroscopy and pH values were determined.

### 2.4. Cell Cultures

Human umbilical vein endothelial cells (HUVEC, Lonza, Basel, Switzerland) were seeded in culture plates coated with 0.2% gelatin. Cultures were maintained in culture medium M199 with 20% of FBS, 2.5 μg/mL fungizone, penicillin-streptomycin (100 IU/mL and 100 μg/mL, respectively), and endothelial cell growth supplement (ECGS, Sigma-Aldrich, Saint Louis, MO, USA). Third-subculture cells were used in the experiments. Cells were seeded at 10^4^ cells/cm^2^ in culture plates coated with 0.2% gelatin. After 24 h, the culture medium was removed, and the cultures were exposed to the extracts from the Mg-based materials for 7 days. The extracts were renewed at medium change (every three days). Cultures were characterized for DNA content, nitric oxide quantification, caspase activity, in vitro angiogenic response, and expression of angiogenic genes. 

Human mesenchymal stem cells (hMSC, Lonza) were expanded in α-minimal essential medium (α-MEM) containing 10% fetal bovine serum (FBS; Gibco/BRL Gaithersburg, MD, USA), 2.5 μg/mL fungizone, penicillin-streptomycin (100 IU/mL and 100 μg/mL, respectively). Third-subculture cells were used in the experiments. Human MSCs were cultured in the presence of 10 mM β-glycerophosphate and 10 nM dexamethasone for the induction of the osteoblastogenic phenotype [27]. Cells were seeded at 10^4^ cells/cm^2^ and after 24 h, the culture medium was removed, and the cultures were treated with the extracts of the Mg-based materials for 21 days. The extracts were renewed at each medium change (once a week). Cultures were accessed for DNA content, ALP activity, and the gene expression of osteoblastogenesis.

Peripheral blood mononuclear cells (PBMCs) were used as the osteoclast precursors. PBMCs were isolated from blood of 25–35 years old healthy male donors after informed consent, as previously described [28]. Shortly, after dilution with phosphate-buffered saline (PBS) (2:1), blood was applied on top of Ficoll-Paque™ PREMIUM (GE Healthcare Bio-Sciences, Pittsburgh, PA, USA). Samples were centrifuged at 400× *g* for 30 min and PBMCs were collected at the interface between Ficoll-Paque and PBS. Cells were washed twice with PBS. On average, for each 90 mL of processed blood, about 450 × 10^6^ PBMC was obtained. PBMC were cultured (1.5 × 10^6^ cells/cm^2^) in in α-MEM supplemented with 30% human serum (from the same donor from which PBMC were collected), 2.5 μg/mL fungizone, penicillin-streptomycin (100 IU/mL and 100 μg/mL, respectively), 2 mM L-glutamine and the osteoclastogenic inducers M-CSF (25 ng/mL), and RANKL (40 ng/mL). After 24 h, the extracts were added, and the cultures exposed for 21 days. The extracts were renewed at medium change (once a week). Cell response was analyzed for DNA content, TRAP activity, and the expression of osteoclastogenesis genes. 

### 2.5. Characterization of the Cellular Response

#### 2.5.1. DNA Content Assay

DNA content assay was performed in all cultures using Quant-iT™ PicoGreen^®^ dsDNA kit (Invitrogen, Molecular Probes, Eugene, OR, USA) according to manufacturer’s instructions. Previously, cultures were treated with 0.1% Triton X-100 for 15 min. Fluorescence was measured on a ELISA reader (Synergy HT, Biotek, Winooski, VT, USA) at wavelengths of 480/520 nm (excitation/emission).

#### 2.5.2. ALP Activity

ALP activity, determined on hMSC cultures, was evaluated in cell lysates (0.1% Triton X-100, 5 min) by the hydrolysis of p-nitrophenyl phosphate in alkaline buffer solution (pH ≈ 10.3; 30 min, 37 °C) and colorimetric determination of the product (p-nitrophenol) at 400 nm in an ELISA plate reader (Synergy HT, Biotek). ALP activity was normalized to total protein content (quantified by Bradford’s method) and was expressed as nmol/min·μg protein. 

#### 2.5.3. TRAP Activity

TRAP activity in PBMC cultures was evaluated in cell lysates (0.1% Triton X-100, 5 min) by the hydrolysis of p-nitrophenyl phosphate in an acidic buffer solution (pH = 5.8, 1 h, 37 °C) and colorimetric determination of the product (p-nitrophenol) at 400 nm in an ELISA plate reader (Synergy HT, Biotek). TRAP activity was normalized to total protein content (quantified by Bradford’s method) and was expressed as nmol/min·μg protein.

#### 2.5.4. Caspase Activity

Caspase activity of HUVEC was performed using EnzCheck caspase-3 assay kit #2 (Invitrogen, Molecular Probes) according to manufacturer’s instructions. Previously, cultures were treated with 0.1% Triton X-100 for 15 min. Fluorescence was measured on a ELISA reader (Synergy HT, Biotek, USA) at wavelengths of 496/520 nm (excitation/emission). Caspase-3 activity was normalized to total protein content (quantified by Bradford’s method) and was expressed as nmol/min·μg protein.

#### 2.5.5. Histochemical Staining of Alkaline Phosphatase

At days 7, 14, and 21 of culture, hMSC cultures were fixed in 1.5% glutaraldehyde in sodium cacodylate (0.14 M) for 10 min. For ALP, cells were rinsed with PBS and stained with a solution of α-napthyl acid phosphate (2 mg/mL) and Fast Blue R (2 mg/mL) prepared in Tris 0.1 M at pH 10. The cultures were incubated for 1 h in the dark and washed with water at the end of the period. The presence of the staining was assessed under a Nikon TMS inverted phase microscope. 

#### 2.5.6. Nitric Oxide (NO) Detection

The presence of NO in HUVEC cultures incubated with Mg-based extracts at 10 and 20% was assessed using a Total Nitric Oxide detection kit (Enzo Life Sciences, Farmingdale, NY, USA). After specific time points, supernatants were collected and processed following the manufacturer’s protocol. The synthesis of NO (µM) was measured at 540 nm using a microplate reader (Synergy HT, Biotek, USA) and values were normalized to total protein content, as previous described. 

#### 2.5.7. Total RNA Extraction and RT-PCR Analysis

Osteoblastic, osteoclastic, and endothelial cell cultures were characterized by RT-PCR for the expression of relevant genes, after 4 (HUVECs) and 14 days (hMSCs and osteoclasts) of exposure to extracts at a concentration of 10%. 

Briefly, total RNA was isolated from cell cultures using Trizol reagent, in accordance with the standard manufacturer’s protocol. RNA’s concentration and purity were assessed by absorbance reading (A260/A280) using the Take3 module (Gen5, BioTek, Winooski, VT, USA) and a microplate reader (Synergy HT; BioTek). The conversion to cDNA was conducted with a reverse transcription system (NZY first-strand cDNA synthesis). Quantitative PCR was performed using a CFX384 real-time PCR system (Bio-Rad, Hercules, CA, USA), with the iTaq Universal SYBR green Supermix PCR Kit and the primers specified in Table 1 (all from Bio-Rad), following the manufacturer’s cycling protocol. Quality control was checked using Bio-Rad CFX Maestro 1.1, v.4.1.24. Only samples with amplification efficiency between 110–90 and a linear standard curve (R^2^) greater than 0.98 were included. The relative quantification of each target gene was normalized using GAPDH levels (the housekeeping gene). Three independent experiments were performed.

#### 2.5.8. Tube-like Formation Assay

The ability of endothelial cell cultures to self-organize the cell layer in a network of tube-like structures upon the contact with an extracellular matrix was evaluated using the Matrigel assay. Endothelial cell cultures were performed as described above. At day 4 of culture, the medium was removed and 10 mg/mL of Matrigel (BD Matrigel Matrix, BD Biosciences, Franklin Lakes, NJ, USA) was added to the culture. After an incubation of 60 min at 37 °C for the jellification of the matrix, culture medium was added to the wells and cultures kept for 24 h. Capillary tube-like formation was assessed under a microscope (Nikon TMS Inverted Phase microscope).

### 2.6. In Vivo Angiogenic Assay—Chick Chorioallantoic Membrane (CAM) Assay

The effect of the extracts from the Mg-based substrates in the in vivo angiogenic response was evaluated by the CAM assay. Fertilized chicken embryos were incubated at 37 °C in 45% humidity for 4 days. After this period, a hole was drilled in order to expose the vascular zone of the embryo. A sterilized filter-paper disk was used as carrier for Mg extracts applied to the vascular zone. The eggs were sealed and incubated for 3 days at 37 °C. After the incubation period, the CAM was photodocumented (Nikon TMS Inverted Phase microscope).

### 2.7. Statistical Analysis

Four independent experiments were performed in all biological assays. In each experiment, three replicas were accomplished for the quantitative assays and two replicas for the qualitative assays. The results are shown as mean ± standard error (±SE). Data were analyzed with one-way analysis of variance (ANOVA). Statistical differences were determined by the post hoc Tukey HSD multi-comparison test. A *p* value of value of <0.05 was considered statistically significant.

## 3. Results and Discussion

### 3.1. SEM Observation of the Mg-Based Samples and Levels of Mg and Ca Ions in the Extracts

Figure 2a shows representative images of the original AZ31 alloy, anodized alloy, and the coated alloy. The grinding marks can be noted on the unpolished surface of the original alloy, as well as some white and black dots, corresponding to aluminum and carbon particles, respectively [29]. In addition, anodization clearly increased the surface’s roughness, with the formation of pores and a net-like structure, due to the presence of oxides on the surface of magnesium substrate. Moreover, the rough surface achieved by the anodization was uniformly covered by the coating process, originating a smooth and homogeneous crack-free surface. The coating layer displayed a dense and non-porous morphology with the presence of HA agglomerates.

The concentrations of Mg and Ca ions and the pH values in the pure extracts from the Mg-based substrates were also evaluated (Figure 2b). Compared to the control (culture medium used to prepare the extracts), Mg levels were higher in the extracts from AZ31, followed by those from anodized AZ31 that were only slightly higher than control, in the extracts from the coated anodized AZ31. These results are expected, being in line with the degradation rate of the three materials that is high in the untreated alloy and very low in the coated anodized alloy, as reported previously [18,19,20,21,22,23]. As expected, from the composition of the materials, the levels of Ca ions in the extracts were similar to those present in the culture media.

pH values followed the same trend as the levels of Mg ions in the extracts. Values were ~9.5, 8.5, and 7.4, respectively in the original, anodized, and coated alloy. Dilutions of the pure extracts from 50% to 2% were used in the cell cultures experiments. In these conditions, the pH of the diluted extracts was 7.4, with the exception of the 50% dilution of the AZ31 and anodized AZ31 sample extracts (pH values = 7.8). Although alkaline conditions disrupt cellular behavior, it is also known that they can recover from mild alkalinity [30]. In this manner, at the dilutions used, the pHs of the extracts are not expected to have any contribution on the elicited cell response.

### 3.2. Effects of the Extracts from Mg-Based Substrates in Endothelial Cells

Following a xenograft-material’s insertion, the healing process starts with the coagulation phase, setting on platelet activation and fibrin clot formation [31]. Right after this early stage, the inflammatory phase begins, promoting a massive angiogenesis at the wound area [32]. This neovascularization allows the removal of debris, providing nutrients and oxygen to the metabolic active wound. During angiogenesis, the appropriate interaction of the material and its leachable/degradation products with endothelial cells is of upmost relevance to the successful accomplishment of the subsequent repair phase. In the present study, endothelial cells were exposed for 4 and 7 days to the extracts from the Mg-based materials at 20% and 10% dilutions. Cell response was characterized for proliferation and apoptosis and, also, for phenotypic markers (NO production, gene expression, and ability to form tubular-like structures). Results are summarized at Figure 3. 

DNA content increased throughout the culture time in all cultures, evidencing a growing cell population (Figure 3a). Compared to control groups, overall, the extracts increase DNA content, although without attaining statistical significance. Caspase-3 activity was also measured in the same cultures (Figure 3b). This protease has a substrate specificity to the amino sequence Asp-Glu-Val-Asp and cleaves several proteins, such has poly(ADP-ribose) polymerase (PARP) and DNA-dependent protein kinase [33], thus having an important role in the initiation of apoptosis. Caspase-3 activity increased slightly from day 4 to day 7 in all cultures, which is related to the higher cell confluence observed at day 7, with an expected increase in cell death by apoptosis. However, compared to the control, the extracts-exposed cultures presented similar values at day 3, although there was a tendency for decreased values at day 7 (more evident with the extracts from the coated anodized AZ31).

Endothelial cell-derived nitric oxide (NO) synthesis is an important marker related to cell survivor. The synthesis of NO, normalized to the protein content, was similar at days 4 and 7 (Figure 3c). Additionally, no significant differences were found between control cultures and those exposed to the extracts. However, at day 7, slightly increased values were noted in the cultures treated with the extracts, which tag along with another studies [34]. 

The expression of the endothelial genes VE-Cadherin, CD31, and VWF of HUVECs was analyzed at day 4. VE-Cadherin plays a major role in the organization of adherent junctions of endothelial cells and can modulate the downstream of growth factor receptors, such as VEGFR2, FGF-R1, and the TGFβ-R [35], while CD31, also known as PECAM-1, is related to the support of the integrity of endothelial cell–cell junctions, in addition to providing protection for the vascular bed to apoptotic stimuli [36]. Von Willebrand factor (VWF) is the master regulator of angiogenesis, endothelial homeostasis, and vascular endothelial growth factor signaling [37]. Results (Figure 3d) showed that all extracts significantly increased the expression of VE-Cadherin and also of CD31 (although statistical significance was only observed with the extracts from coated AZ31). The expression of VWF was comparable in all cultures.

The in vitro ability to organize the cell layer in a network of cord-like formations upon contact with an extracellular matrix is an important feature of endothelial cells, suggesting its functional performance in undergoing angiogenesis. Figure 4a shows that all cultures presented this feature, and no differences were noted between the control cultures and those exposed to the extracts. It is known that sprouting angiogenesis depends on endothelial cell migration, which is related to increased mitochondrial and metabolic activity [38]. In the present work, the CAM assay showed that after three days-exposure to the undiluted extracts (from the three Mg-based materials), angiogenesis was observed in all conditions in a manner similar to the control (Figure 4b). 

It is known that Mg ions can modulate the proliferation and function of endothelial cells by several mechanisms, and the effects are dose-dependent [38]. Considering the concentration of Mg ion in the pure extracts, in the present study, endothelial cells were exposed to ~3.4 mM (20% dilution) and ~1.7 mM (10% dilution) Mg ion present in the extract from AZ31 and lower levels with the other extracts. Overall, at these concentrations, slightly increased metabolic activity was observed and also the higher expression of VE-Cadherin; the expression of CD31 was also upregulated with the extract from the coated anodized AZ31 alloy. However, functional markers such as NO production, in vitro formation of cord-like structures, and in vivo angiogenesis were not affected. Previous studies reported variable results. Thus, human umbilical cord perivascular cells exposed to Mg ionic concentrations around 10 mM presented improved proliferation and migration, higher total cellular actin, and an upregulation of NO synthase III and cadherin-5 [39]. Increased proliferation was found also in human coronary artery endothelial cells exposed to 10 mM Mg ion [40]. However, in another study, HUVEC showed faster proliferation rate, higher NO production, higher levels of angiogenic-related gene expression, and greater cell area when exposed to Mg-extracts containing a Mg concentration of 2.31 mM, compared to exposure to those having 5.25 and 6.46 mM [34]. Regarding functional activity, HUVEC incubated with Mg extracts (4 to 8 mM) and under normoxia conditions exhibited VEGFB upregulation and enhanced migration but a decreased tubule formation [38]. However, Mg ion concentrations up to 10 mM have been described to increase the migration rate of endothelial cells and angiogenesis [39,41]. One possible explanation for these discrepancies is due to the presence and the ionic release of other elements present in the alloys, which can produce toxic effects depending on local concentrations once some studies were performed using diverse Mg-based alloys instead of pure Mg or Mg salt solutions [39].

### 3.3. Effects of the Extracts from Mg-Based Substrates in Bone Cells

Mg has a vital role in several biological functions, as the maintenance of membrane integrity and the regulation of cell proliferation, differentiation, and apoptosis [42,43,44]. Additionally, Mg acts directly in the activity of osteoclasts and osteoblasts, regulating bone turnover and mineral homeostasis. In this work, osteoblastic-differentiating mesenchymal stem cells and osteoclastic-differentiating PBMCs were exposed to the extracts from the Mg-based substrates (50–2% dilutions) for 21 days. In these conditions, cells contacted with Mg ion concentrations of ~11.5–0.42% mM (AZ31), ~6.5–0.26% mM (anodized AZ31), and ~1 mM (coated AZ31, the Mg concentration present in the culture medium). Cell response was characterized for DNA content and specific phenotype markers.

#### 3.3.1. Osteogenic-Differentiating Mesenchymal Stem Cells

Figure 5 summarizes the results observed in hMSC cultured in osteogenic conditions (presence of dexamethasone and beta-glycerophosphate) [27] and exposed to the extracts of Mg-based samples. In control conditions, cells proliferated throughout the 21-day cultures, as shown by the DNA values. The main relevant effects of the extracts (Figure 5a) were observed with those from AZ31 and anodized AZ31, i.e., slightly decreased values with the 50% dilution but a small increase with the 10 and 5% dilutions. The extract from the coated AZ31 did not affect DNA content. ALP activity followed a similar trend (Figure 5b). Compared to the control, the most relevant effect was the higher enzyme activity at day 14 in the cultures treated with AZ31 and anodized AZ31 extracts at 10 and 5% dilutions. This stimulatory effect was well evident in the cultures stained for the presence of ALP exposed to the extracts at 10% dilution (Figure 5c). These cultures presented an increase in ALP staining at days 14 and 21 compared to the control. In addition, cells presented a characteristic pattern of cell growth, exhibiting areas of higher cell density randomly distributed on the culture surface, suggesting an induced osteoblastic differentiation. Otherwise, the extracts from the coated alloy caused no significant effects on ALP activity (Figure 5b) and staining (Figure 5c). The gene expression of osteoblast-related markers was evaluated in cultures exposed to 10% dilution extracts (Figure 5d). Experimental groups showed a significant increase in the expression of ALP, Runx-2, and OPG in comparison to the control, while no significant differences were observed in Col-1, BMP-2, and osteocalcin. 

MSCs have a key role in biomaterials-mediated bone regeneration/healing. During the proliferation/differentiation pathway, the initial fast proliferation rate intended to increase the cell population through cell cycle events, mainly DNA synthesis and cell division. Ongoing osteoblastic differentiation is associated with a decrease in the proliferation rate while increasing osteoblastic features [45]. In the present study, control hMSC cultured in the presence of dexamethasone and beta-glycerophosphate followed this pattern (Figure 5). Cells produced ALP, an early osteoblastic marker with an important role in the initiation of the matrix mineralization [27] and expressed the genes coding for Runx2 (the master regulator of the osteoblastic differentiation), ALP, collagen type 1 (the main component of the extracellular matrix), BMP-2, OCN (typical osteoblastic genes), and OPG (with a key role in the interaction with the osteoclastic cells in the bone microenvironment). The extracts from the Mg-based substrates had an effect in this behavior, which is clearly dependent on the different compositions of the extract yielded from the three Mg-based materials due to the differences in the degradation rate. Regarding the levels of Mg ions, in the tested concentration range (50–2% dilutions), cells were exposed to concentrations of ~11.5 to ~1 mM, considering the three tested substrates. 

Extracts from AZ31 and anodized AZ31 at the higher concentrations had a negative effect on cell proliferation and ALP activity but increased these parameters at lower concentrations (~10% dilution). The lowest concentration (2% dilution) had no significant effect. The same was observed with the extracts from the coated AZ31 (with Mg ion levels similar to those in the culture medium). These observations are corroborated by a variety of previous studies. It has been widely reported that the released Mg ions during material degradation regulated gene and protein expression associated with MSCs growth and differentiation, and its effects were highly dose-dependent [46,47]. A variety of works exposing the cells to different ranges of Mg ion concentrations, prepared from the dissolution of Mg salts or present in extracts from Mg-based materials, has been reported attempting to estimate an optimized value. Interestingly, a variety of in vitro studies conducted with MSCs and osteoblastic cell lines converged to the concentration range of 2–10 mM Mg ion for enhancing cell metabolism and upregulating proliferation and early osteoblastic differentiation rates [48,49,50]. By contrast, higher Mg concentrations appeared cytotoxic [51,52]. Of note, a recent reported computer model, calibrated using experimental data, showed that Mg ions within the range of 3–6 mM increased proliferation and early differentiation, whereas high levels caused deleterious effects [53].

#### 3.3.2. Osteoclastogenic-Differentiating Mononuclear Precursor Cells

In control conditions, the mononuclear osteoclastic precursors of the peripheral blood were cultured in the presence of the factors MCSF and RANKL, known to induce osteoclastogenesis in vitro [54]. During osteoclastogenic differentiation, the high number of seeded mononuclear cells fused to form a much lower number of multinucleated functional osteoclasts at longer incubation times [55]. Accordingly, in the present work (Figure 6), DNA content decreased with culture incubation (Figure 6a), while TRAP activity increased (Figure 6b), reflecting an osteoclastogenic differentiation process. Additionally, at day 14, cultures expressed a set of osteoclastogenic genes (Figure 6c), i.e., coding for C-src, TRAP, calcitonin receptors (specific of osteoclasts cells), and Catk, a protease involved in the resorption process [56]. 

Exposure to the extracts from Mg-based materials clearly affected this behavior. Extracts from the uncoated samples (Figure 6a), at high concentrations (50 to 10%), caused an increase in DNA content at day 7 (AZ31) and days 7 and 14 (anodized AZ31), followed by a significant decrease at longer culture times (50 and 20%). Higher extract dilutions did not affect DNA content. Similarly to that observed with the osteoblastic cells, the extracts from the coated AZ31 did not affect DNA values (Figure 6a). Effects on TRAP activity (Figure 6b) were mainly noted at day 14 and included an increase in enzyme activity with exposure to low extract concentrations (10 and 5%) from AZ31 and anodized AZ31. Extracts (10%) from the same materials also induced the expression of genes coding for TRAP and calcitonin receptors (Figure 6c). By contrast, and similarly to that observed with DNA content, the extracts from the coated alloy did not affect the osteoclastic differentiation behavior (TRAP activity and gene expression, Figure 6b,c).

Some previous in vitro studies also addressed the effect of Mg ions and/or extracts from Mg-based materials in different osteoclastic cell culture models [30,57,58,59]. The reported results show great variability concerning early and late osteoclastic differentiation and cell functionality. Contributing factors include differences in tested materials, preparation of the extracts (material/medium ratio and non-filtered/filtered extracts), culture models, cell differentiation stage, Mg ion concentration ranges, and pH of the extract, among others [57]. Similarly to that described for the modulation of osteoblastic behavior by Mg ions/material extracts, the effects on osteoclastic lineage cells are highly dependent of the Mg ion concentration, but the reported discrepancy does not allow a direct result comparison.

Nevertheless, in the present work and according to the proposed aim, results clearly evidence that the extracts from the coated AZ31 alloy (the final substrate) did not interfere with the behavior of the osteoclastic differentiation tested parameters. As mentioned above, this extract contains a very low Mg ion concentration, i.e., similar to that present in the culture medium, due to the very low degradation rate as shown in previous studies [18,19,20,21,22,23]. This was not the case of the extracts from anodized AZ31 and, in particular, the original alloy (Figure 2b). Higher extract dilutions (10% and 5%) did cause a stimulatory effect in TRAP activity, an enzyme beginning to be expressed early in the differentiation pathway and still present in the mature osteoclast [55], along with an up-regulation of the genes coding for TRAP (as expected, considering the higher enzyme activity) and for calcitonin receptors, a specific marker of these cells [54,56]. Nevertheless, it is worth noting that the expression of genes for the later markers C-src, with a key role in osteoclast bone resorbing activity, and Catk, a protease involved in the degradation of type I collagen in the bone matrix were not affected, probably suggested that the tested extracts did not induce the formation of fully mature resorbing osteoclasts compared to the control. 

### 3.4. Integrating the Cell Response to the Extracts from the Mg-Based Substrates

Mg-based materials degrade progressively at the implantation site, which is associated to changes in the local environment, namely concerning ion composition, pH increase, and the presence of material particle/debris, affecting a range of processes involved in tissue regeneration [47,60]. Surface treatments to tailor the degradation rate to the tissue regeneration kinetics are difficult tasks and are continually sought. In previous studies, we reported the development and characterization of a degradable multifunctional platform by conjoining an anodization treatment on AZ31 alloy followed by a multifunctional polymeric coating reinforced with 2% of nanohydroxyapatite [18,19,20,21,22,23]. This platform elicited enhanced bone regeneration upon tibia implantation within a rabbit model [22], but the cellular and molecular underlying mechanisms were not disclosed. Now, the present study attempted to simulate in vitro the bone repairing effect of this degradable platform. This was accomplished by exposing relevant cells in biomaterials-mediated bone regeneration, i.e., endothelial cells and bone lineage cells, to the extracts from the multifunctional platform (the implantable material) as well as the underlying anodized and original AZ31 alloy.

Following implantation, local fluids and cells immediately interact with the multifunctional platform, and the established inflammatory process triggers an active angiogenesis process essential for the onset and ongoing repair phase by supporting the vascularization during all regeneration processes [61,62]. Thus, the interactions of the material surface and the degradation products with the endothelial cells were observed, which were soon followed by osteoblastic precursors, starting the repair phase, and the osteoclastic cells for the remodeling events [61,62]. Through the regeneration process, the material is degraded and local cells first contact with the degradation products from the coated alloy and, then, progressively, with those from the anodized and the original alloy. The present results showed that the extracts from the coated AZ31 platform had little effect on the three analyzed cell types, which is expected due to the very slow degradation rate of this material [19,20], as reflected by the levels of Mg ions similar to those on the culture medium. Thus, in the initial stages of the bone regeneration events, relevant local cells interact with a cytocompatible material surface, as demonstrated previously [21], and, furthermore, they are not affected by the eventual degradation products (Figure 3, Figure 5 and Figure 6). Nevertheless, it is interesting to note that the extracts, at appropriate concentrations, had a small inductive effect on the proliferation and gene expression of endothelial cells (Figure 3) and also in some osteoblastic genes (Figure 5), but no effect was observed in osteoclastic cells (Figure 6), suggesting the possibility of a slight induction of the early repair events. However, during the progressive degradation of the polymeric coating, local cells will interact with the degradation products from the anodized and original alloy, which have a higher degradation rate, as seen previously [18,19,20,21,22,23], and as it is evidenced by the levels of Mg ions in the respective extracts (Figure 2b). Results suggested that, at appropriate levels, in vivo effects will occur in some stage of the degradation kinetics, and the leaching products from these substrates are compatible with an inductive effect in both osteoblastic- and osteoclastic-differentiating cells, as predictable by the resulting higher ALP and TRAP activities and gene expression of some phenotype genes in both cell types (Figure 5 and Figure 6). The presence of Mg ions, within a defined concentration range, appears to have a contribution on these effects. However, it appears from the results of the gene expression profile that this inductive effect might be occurring at early stages of bone cells differentiation. Still, the integration of these observations suggest the possibility of increased bone metabolism dynamics contributing to both a faster bone formation and remodeling rates. Overall, the results of the present study, detailing the effects of the leachable products from the coated AZ31 alloy (the implantable material) and from the underlying substrates during the progressive degradation of the implanted material will contribute to explain the enhanced in vivo bone regeneration potential of the multifunctional coated AZ31 platform, as seen previously [22]. Figure 7 summarizes the hypothesized mechanisms.

## 4. Conclusions

This study intended to elucidate, in an in vitro approach, the cellular and molecular mechanisms occurring during bone regeneration upon the implantation of a tailored degradable multifunctional Mg-based alloy, prepared by conjoining anodization of AZ31 alloy followed by the deposition of a functionalized polymeric coating. Human endothelial cells and osteoblastic and osteoclastic differentiation cells were exposed to the extracts from the multifunctional platform (the implantable material, having a low degradation rate), as well as the underlying anodized and original AZ31 alloy (with higher degradation rates). Extracts from the polymeric coated alloy did not affect cellular behavior, although a small inductive effect was observed in the proliferation and gene expression of endothelial and osteoblastic cells. Extracts from the higher degradable anodized and original alloy, within an appropriate concentration range, induced some endothelial genes and, also, ALP and TRAP activities, further increasing the expression of some early differentiation osteoblastic and osteoclastic genes. The concentration range of Mg ions present in the extracts appeared to contribute to the observed inductive effects. The integration of these results with previous outcomes of in vitro direct assays where osteoblastic cells were found to adhere and actively proliferate over the coating’s surface suggests that, following the implantation of a tailored degradable Mg-based material, the absence of initial deleterious effects would favor the early stages of bone repair and, subsequently, the on-going degradation of the coating and subjacent alloy would increase bone metabolism dynamics favoring faster bone formation and remodeling processes, corroborating the results of previous in vivo implantation. 

## Figures and Tables

**Figure 1 bioengineering-09-00255-f001:**
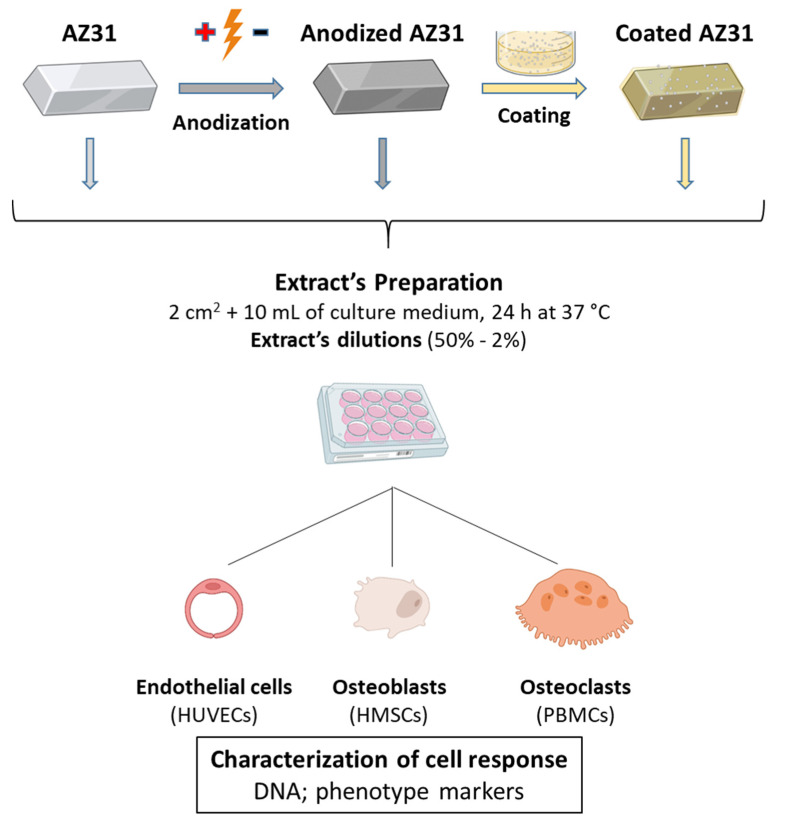
Diagram of the experimental protocol used to evaluate the effects of the extracts from the Mg-based substrates on relevant cells involved in bone regeneration.

**Figure 2 bioengineering-09-00255-f002:**
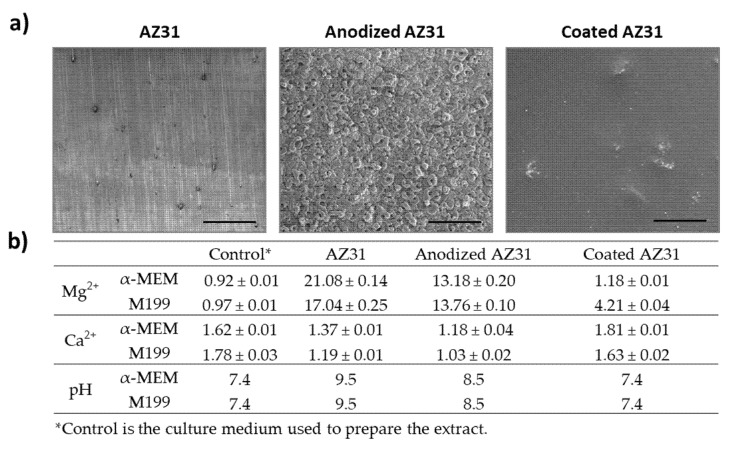
(**a**) Representative SEM images of the Mg-based substrates. Bar: 100 µm. (**b**) Concentration of Mg and Ca ions in the pure extracts (mM).

**Figure 3 bioengineering-09-00255-f003:**
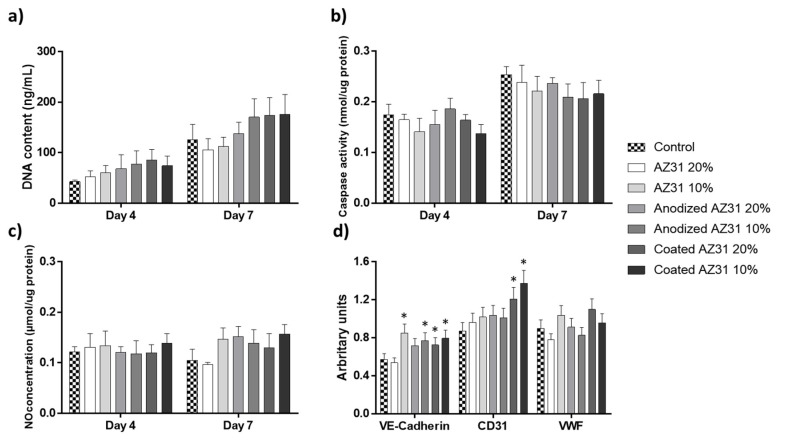
DNA content (**a**), Caspase-3 activity (**b**), NO concentration (**c**), and gene expression (**d**) of HUVECs cultured in the presence of the extracts from Mg-based materials and 20% and 10% dilutions, at days 4 and 7. * Significantly different from control (absence of the extracts).

**Figure 4 bioengineering-09-00255-f004:**
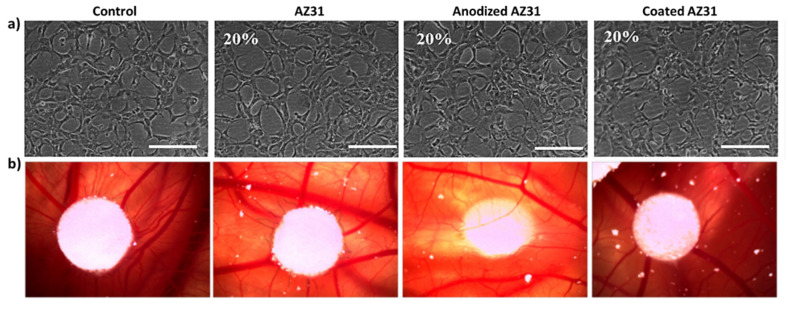
(**a**) The Matrigel assay: representative images of HUVECs cultured in control conditions and in the presence of the extracts from the Mg-based materials (20%), showing the organization of the cell layer in a network of cord-like structures upon the addition of Matrigel. Scale bar: 200 μm. (**b**) The CAM assay: effect of the extracts from the Mg-based alloys in the angiogenic response. Representative images of the angiogenic response surrounding the filter samples impregnated with the extracts.

**Figure 5 bioengineering-09-00255-f005:**
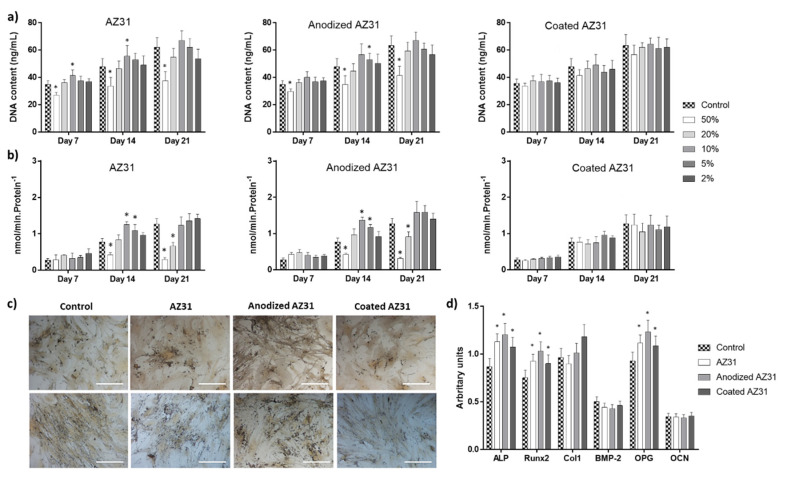
Human MSCs cultures exposed to the extracts from the Mg-based substrates (AZ31, anodized AZ31 and coated AZ31), at days 7, 14 and 21. (**a**) DNA content; (**b**) ALP activity; (**c**) ALP staining; (**d**) expression of osteoblastic genes. * Significantly different from control cultures (absence of the extracts).

**Figure 6 bioengineering-09-00255-f006:**
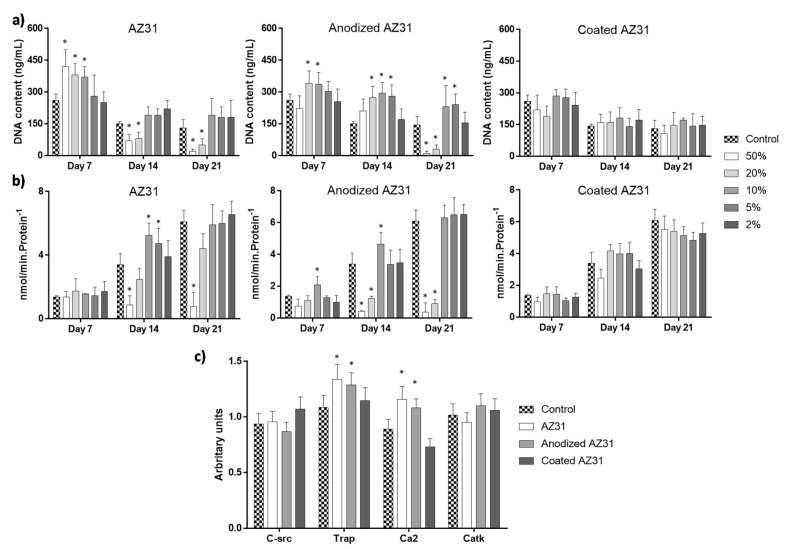
Human PBMCs cultures exposed to the extracts from the Mg-based substrates (AZ31, anodized AZ31 and coated AZ31), at days 7, 14, and 21. (**a**) DNA content; (**b**) TRAP activity; (**c**) expression of osteoclastic genes. * Significantly different from control cultures (absence of the extracts).

**Figure 7 bioengineering-09-00255-f007:**
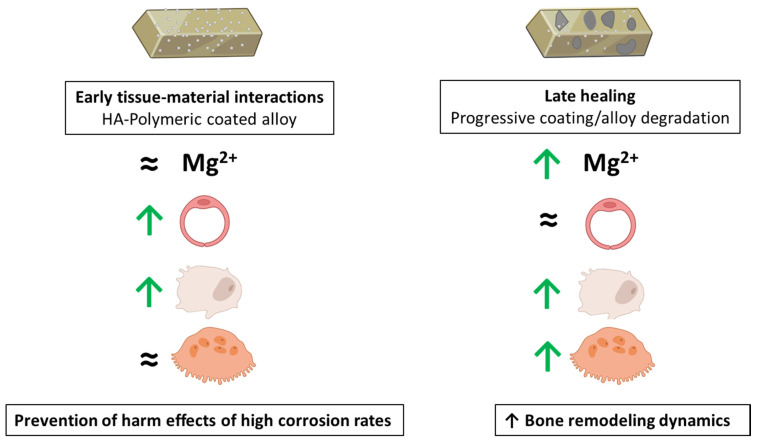
The main effects of the multifunctional Mg-based alloy on HUVECs, hBMCs, and osteoclasts. ≈ No differences in comparison to the control cultures (absence of the extracts).

**Table 1 bioengineering-09-00255-t001:** The unique assay ID of the used primers purchased from Bio-Rad.

Gene	Assay ID
Housekeeping gene	
GAPDH	qHsaCED0038674
Osteoblastic genes	
ALP	qHsaCED0045991
Runx-2	qHsaCED0044067
Col-1	qHsaCED0043248
BMP-2	qHsaCID0015400
OPG	qHsaCED0046251
OCN	qHsaCED0038437
Osteoclastic genes	
C-src	qHsaCID0011233
TRAP	qHsaCED0056724
Ca2	qHsaCID0021039
Catk	qHsaCID0016934
Endothelial genes	
VE-cadherin	qHsaCID0016288
CD31	qHsaCED0045459
VWF	qHsaCED0033955

Abbreviations: GAPDH: Glyceraldehyde-3-Phosphate Dehydrogenase. ALP: Alkaline Phosphatase; Runx-2: Runt-Related Transcription Factor 2; Col-1: Collagen Type I Alpha 1; BMP-2: Bone Morphogenetic Protein 2; OPG: Osteoprotegerin; OCN: Osteocalcin. C-src: SRC Proto-Oncogene; Non-Receptor Tyrosine Kinase; TRAP: acid phosphatase 5 tartrate resistant; Ca2: Carbonic Anhydrase 2; Catk: Cathepsin K. VE- cadherin: vascular endothelial cadherin; CD31: Platelet and Endothelial Cell Adhesion Molecule 1; VWF: Von Willebrand Factor.

## Data Availability

The data presented in this study are available upon request from the corresponding author.

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
