# Peer review of "Simulating In Vitro the Bone Healing Potential of a Degradable and Tailored Multifunctional Mg-Based Alloy Platform"

_bioengineering, 2022, doi:10.3390/bioengineering9060255_

Round 1
Reviewer 1 Report
The manuscript describes the preparation of functional surfaces on Mg-based metal alloys and the effects on vascularization, bone formation and bone resorption. This is well written and is likely to be of significant interest to the readers of Bioengineering. However, the descriptions of study design, results and discussions are not sufficient. The following concerns or suggestions should be addressed before acceptance of this manuscript. Then this manuscript is acceptable for publication on the journal with minor revision.
1. The authors mentioned “In this context, magnesium-based (Mg-based) metal alloys have taken the frontline of a new generation of biodegradable metal-based materials with promising bone-related properties including: density, elastic modulus and compressive yield strength closer to the values observed in natural bone [10]” in the section of introduction. However, the fact is that some of the Mg based metal alloys have different mechanical properties from bone, for instance, yield stress (MPa) and elastic modulus (GPa). The authors are recommended to provide correct information.
2. The functional surfaces of Mg based metal alloys were prepared by coating with nano-sized hydroxyapatite and composite polymer. The authors are recommended to provide the information of the coating materials more in details, for instance, particle size, amount, density of the nano-sized hydroxyapatite and thickness of the polymer coating.
3. The molecular markers for osteoclast differentiation the authors used in this study are common in osteoclasts and foreign body giant cells who share several characteristics. An important distinction between these cell types is that osteoclasts can resorb bone, but foreign body giant cells cannot. The authors are recommended to show the results of resorption.
4. What is the reason why the authors selected the time point of 24 h for preparation of extracts in cell culture medium? The biomaterials might stay in body for longer time.
Author Response
The manuscript describes the preparation of functional surfaces on Mg-based metal alloys and the effects on vascularization, bone formation and bone resorption. This is well written and is likely to be of significant interest to the readers of Bioengineering. However, the descriptions of study design, results and discussions are not sufficient. The following concerns or suggestions should be addressed before acceptance of this manuscript. Then this manuscript is acceptable for publication on the journal with minor revision.
- The authors mentioned “In this context, magnesium-based (Mg-based) metal alloys have taken the frontline of a new generation of biodegradable metal-based materials with promising bone-related properties including: density, elastic modulus and compressive yield strength closer to the values observed in natural bone [10]” in the section of introduction. However, the fact is that some of the Mg based metal alloys have different mechanical properties from bone, for instance, yield stress (MPa) and elastic modulus (GPa). The authors are recommended to provide correct information.
Response: The authors thank the reviewer for the observation. The sentence has been corrected:
“In this context, magnesium-based (Mg-based) metal alloys have taken the frontline of a new generation of biodegradable metal-based materials with promising bone-related properties. For instance, these materials present similar values of density, tensile strength and compressive yield strength to cortical bone [10]. Despite other properties as yield stress and elastic modulus are yet distinct, Mg-based materials are the best contender to replace the non-resorbable solutions for bone fixation.”
- The functional surfaces of Mg based metal alloys were prepared by coating with nano-sized hydroxyapatite and composite polymer. The authors are recommended to provide the information of the coating materials more in details, for instance, particle size, amount, density of the nano-sized hydroxyapatite and thickness of the polymer coating.
Response: The authors thank the reviewer for the suggestion. The information was added within the 2.1 materials preparation section:
“Coated anodized AZ31 samples. The polymeric coating was synthesized by mixing PEI in N,N-dimethylacetamide (DMAc), as a solvent, in concentrations of 15 wt.%. The mixture was stirred for 24 h at 50 ºC in order to obtain a stable and yellowish but transparent solution. Diethylenetriamine (DETA) (3 wt. %) was added to this formulation. To improve the biocompatibility, nano-hydroxyapatite particles (HA) with 15-20 nm of particle size were added to the obtained polymer (2% wt.). The composite polymeric coating (PEI+DETA+2%HA) was applied on the anodized coupons using a dip coater, being following cured in an oven at 150 ºC for 2 h under atmospheric conditions, obtaining a coating thickness of ~ 5µm. Detailed description of the methodology was previously reported [18]”.
- The molecular markers for osteoclast differentiation the authors used in this study are common in osteoclasts and foreign body giant cells who share several characteristics. An important distinction between these cell types is that osteoclasts can resorb bone, but foreign body giant cells cannot. The authors are recommended to show the results of resorption.
Response: The authors thank the reviewer for raising the question. Regarding the differentiation procedure, despite that M-CSF is used to differentiate monocytes into both cell types (osteoclasts and FBGCs), RANKL is an exclusive inducer of the osteoclastic differentiation. In order to obtain FBGCs, monocytes have to receive interleukins (e.g. IL-4 and IL-13), which were not employed in the present study. In addition, Cathepsin K was expressed in high amounts by the human PBMCs cultures, which is relatively specific for osteoclasts, once FBGCs express very limited amounts of it. Moreover, the present study was more focused on the genetics mechanisms of the osteoclastic differentiation provoked by the leachates. For an in-depth osteoclasts’ activity analysis (including bone resorption), in vitro direct assays or in vivo implantations would be recommended.
- What is the reason why the authors selected the time point of 24 h for preparation of extracts in cell culture medium? The biomaterials might stay in body for longer time.
Response: The authors thank the reviewer for raising the question. The time point (24h) was chosen due to be the peak of the pro-inflammatory response after a fracture or any bone-related surgery occur. In addition, Type H vessels (described as a subtype of blood vessel in bone that play critical roles in the modulation of bone formation and repair) are found to appear after 24 h, reinforcing the importance of this time point, especially for an endothelial evaluation. This information was added in the manuscript (section 2.3 Extract preparation).
Reviewer 2 Report
Magnesium alloy combined with cellular components is an effective medical method for rapid repair of bone tissue. The concentration of calcium and magnesium ions should be marked in Fig 2-b. It is Mg2+ or Ca2+becouse the alloy was anodized.
VWF gene expression is not required in Fig. 3 d.
The use of image of bones is inappropriate in Fig. 7 because it is a simulation experiment.
Regarding “Integration of these results in a translational approach suggests that, following implantation of a tailored degradable Mg-based material, the absence of initial deleterious effects would favor the early stages of bone repair and, subsequently, the on-going degradation of the coating and the subjacent alloy would increase bone metabolism dynamics favoring a faster bone formation and remodeling process”, it is inappropriate to depart from the results of the paper.
Author Response
Reviewer 2
- Magnesium alloy combined with cellular components is an effective medical method for rapid repair of bone tissue. The concentration of calcium and magnesium ions should be marked in Fig 2-b. It is Mg2+ or Ca2+becouse the alloy was anodized.
Response: The authors thank the reviewer for the observation. The table was corrected, with the addition of 2+ with Mg and Ca symbols.
- VWF gene expression is not required in Fig. 3 d.
Response: The authors thank the reviewer for the suggestion. Despite that no significant differences of VWF gene expression were detected, the authors think that the maintenance of the results in the graph might help with the line of thinking of the mentioned assay.
- The use of image of bones is inappropriate in Fig. 7 because it is a simulation experiment.
Response: The authors thank the reviewer for the suggestion. The bone has been removed from the image.
- Regarding “Integration of these results in a translational approach suggests that, following implantation of a tailored degradable Mg-based material, the absence of initial deleterious effects would favor the early stages of bone repair and, subsequently, the on-going degradation of the coating and the subjacent alloy would increase bone metabolism dynamics favoring a faster bone formation and remodeling process”, it is inappropriate to depart from the results of the paper.
Response: The authors thank the reviewer for the suggestion. The phrase was modified:
“Integration of these results with previous outcomes of in vitro direct assays where osteoblastic cells were found to adhere and actively proliferate over the coating’s surface [21], suggests that, following implantation of a tailored degradable Mg-based material, the absence of initial deleterious effects would favor the early stages of bone repair and, subsequently, the on-going degradation of the coating and the subjacent alloy would increase bone metabolism dynamics favoring a faster bone formation and remodeling process, corroborating the results of the previous in vivo implantation [22].”
Reviewer 3 Report
The manuscript is very well written and can be accepted after English language editing.
Author Response
Reviewer 3
The manuscript is very well written and can be accepted after English language editing.
Response: The authors thank very much the reviewer for the approbation. The English language has been revised.
Reviewer 4 Report
The manuscript was well organized and clearly written. I just have two questions regarding the test methods.
(1) The authors used the extracts of Mg-based materials to test cell activity, osteoinduction, and angiogenesis. How did the Mg2+ concentrations in the extracts correlate with in vivo degradation of the alloy? do the authors have in vitro degradation kinetics data of the alloys?
(2) The authors applied coatings to functionalize the surfaces. Do the authors have data on the thickness, homogeneity, and roughness of the coatings? Could the authors exclude the influence from surface properties from experiments? (the osteoinducaiton, angiogenesis are described as a results from Mg2+).
Author Response
Reviewer 4
The manuscript was well organized and clearly written. I just have two questions regarding the test methods.
(1) The authors used the extracts of Mg-based materials to test cell activity, osteoinduction, and angiogenesis. How did the Mg2+ concentrations in the extracts correlate with in vivo degradation of the alloy? do the authors have in vitro degradation kinetics data of the alloys?
Response: The authors thank the reviewer for raising the question. The concentration range (50, 20, 10, 5 and 2%) was set on the dilution of concentrated extracts, in order to cover the variation of materials’ degradation during their interaction with biological systems. The in vitro corrosion resistance of these materials were evaluated in previous studies [18], as well as the implants’ volume loss after their in vivo insertion [22].
(2) The authors applied coatings to functionalize the surfaces. Do the authors have data on the thickness, homogeneity, and roughness of the coatings? Could the authors exclude the influence from surface properties from experiments? (the osteoinducaiton, angiogenesis are described as a results from Mg2+).
Response: The authors thank the reviewer for raising the question. In order to provide a detailed specification of the coating, data (e.g. thickness and HA particle size) were added within the 2.1 materials preparation section. In the present study, the influence of the surface was not addressed, once only the material’s leachates were assessed. However, a detailed analysis focused on the surface’s influence was performed in previous studies using in vitro direct assays, where the cells were seeded over the material’s surface [18].